# Anti-PD-1 Therapy in Advanced Pediatric Malignancies in Nationwide Study: Good Outcome in Skin Melanoma and Hodgkin Lymphoma

**DOI:** 10.3390/cancers16050968

**Published:** 2024-02-28

**Authors:** Agata Marjańska, Katarzyna Pawińska-Wąsikowska, Aleksandra Wieczorek, Monika Drogosiewicz, Bożenna Dembowska-Bagińska, Katarzyna Bobeff, Wojciech Młynarski, Katarzyna Adamczewska-Wawrzynowicz, Jacek Wachowiak, Małgorzata A. Krawczyk, Ninela Irga-Jaworska, Jadwiga Węcławek-Tompol, Krzysztof Kałwak, Małgorzata Sawicka-Żukowska, Maryna Krawczuk-Rybak, Anna Raciborska, Agnieszka Mizia-Malarz, Agata Sobocińska-Mirska, Paweł Łaguna, Walentyna Balwierz, Jan Styczyński

**Affiliations:** 1Department of Pediatric, Hematology and Oncology, Jurasz University Hospital, Collegium Medicum, Nicolaus Copernicus University Toruń, 85-094 Bydgoszcz, Poland; agata.marjanska@wp.pl; 2Department of Pediatric, Oncology and Hematology, Jagiellonian University Medical College, 30-663 Cracow, Poland; katarzyna.pawinska-wasikowska@uj.edu.pl (K.P.-W.); a.wieczorek@uj.edu.pl (A.W.); walentyna.balwierz@uj.edu.pl (W.B.); 3Department of Oncology, The Children’s Memorial Health Institute, 04-730 Warsaw, Poland; m.drogosiewicz@ipczd.pl (M.D.); b.dembowska@ipczd.pl (B.D.-B.); 4Department of Pediatrics, Oncology and Hematology, Medical University of Łodz, 91-738 Łodz, Poland; katarzynabobeff@gmail.com (K.B.); wojciech.mlynarski@umed.lodz.pl (W.M.); 5Department of Pediatric Oncology, Hematology and Transplantology, Jonscher Clinical Hospital, Marcinkowski University of Medical Sciences in Poznań, 60-572 Poznań, Poland; katarzynaadamczewska@wp.pl (K.A.-W.); wachowiak.jacek@outlook.com (J.W.); 6Department of Pediatrics, Hematology and Oncology, Medical University of Gdańsk, 80-210 Gdańsk, Poland; malgorzata.krawczyk@gumed.edu.pl (M.A.K.); ninela.irga-jaworska@gumed.edu.pl (N.I.-J.); 7Department of Bone Marrow Transplantation, Pediatric Oncology and Hematology, Mikulicz-Radecki University Clinical Hospital, 50-556 Wrocław, Poland; jadwiga.weclawek@gmail.com (J.W.-T.); krzysztof.kalwak@gmail.com (K.K.); 8Department of Pediatric Oncology and Hematology, Medical University of Białystok, 15-274 Białystok, Poland; malgorzata.sawicka-zukowska@umb.edu.pl (M.S.-Ż.); maryna.krawczuk-rybak@umb.edu.pl (M.K.-R.); 9Department of Oncology and Surgical Oncology for Children and Youth, Institute of Mother and Child, 01-211 Warsaw, Poland; anna.raciborska@hoga.pl; 10Department of Pediatric, Oncology, Hematology and Chemotherapy, Upper Silesia Children’s Care Health Centre, Medical University of Silesia, 40-752 Katowice, Poland; amizia-malarz@sum.edu.pl; 11Department of Oncology, Children’s Hematology, Clinical Transplantology and Pediatrics, University Clinical Center, Medical University of Warsaw, 02-091 Warsaw, Poland; asmirska@o2.pl (A.S.-M.); pawel.laguna@uckwum.pl (P.Ł.)

**Keywords:** nivolumab, pembrolizumab, ipilimumab, anti-PD-1, immune checkpoint inhibitor, pediatric cancer, malignant melanoma, Hodgkin lymphoma, neuroblastoma

## Abstract

**Simple Summary:**

The aim of this study was to analyze the safety and effectiveness of ICIs used in 42 pediatric patients with various types of highly advanced malignancies. Good outcomes were achieved in patients with malignant cutaneous melanoma or Hodgkin lymphoma. Age > 14 years and good performance status were favorable prognostic factors.

**Abstract:**

Background/aim: The role of immune checkpoint inhibitors (ICIs; anti-PD1) in the treatment of childhood cancers is still evolving. The aim of this nationwide retrospective study was to assess the safety and effectiveness of ICIs used in a group of 42 patients, with a median age of 13.6 years, with various types of advanced malignancies treated in pediatric oncology centers in Poland between 2015 and 2023. Results: The indications for treatment with anti-PD1 were as follows: Hodgkin lymphoma (11); malignant skin melanoma (9); neuroblastoma (8); and other malignancies (14). At the end of follow-up, complete remission (CR) was observed in 37.7% (15/42) of children and disease stabilization in 9.5% (4/42), with a mean survival 3.6 (95% CI = 2.6–4.6) years. The best survival (OS = 1.0) was observed in the group of patients with Hodgkin lymphoma. For malignant melanoma of the skin, neuroblastoma, and other rare malignancies, the estimated 3-year OS values were, respectively, 0.78, 0.33, and 0.25 (*p* = 0.002). The best progression-free survival value (0.78) was observed in the group with malignant melanoma. Significantly better effects of immunotherapy were confirmed in patients ≥ 14 years of age and good overall performance ECOG status. Severe adverse events were observed in 30.9% (13/42) patients.

## 1. Introduction

Every year, approximately 400,000 cases of cancer are diagnosed in children and adolescents worldwide. Over the years, significant progress has been made in therapy. The current effectiveness of anti-cancer therapy is estimated to exceed 80% [1]. However, there are still malignancies in which the efficacy of therapy remains unsatisfactory. This category includes malignant bone tumors, high- and very-high-risk soft tissue sarcomas, selected brain tumors, and high-risk neuroblastoma [2,3]. Additionally, a challenge in therapy involves patients who do not respond to treatment, as well as those with disease progression and recurrence. Hence, there is a continuous need to explore new therapeutic options, with immunotherapy being one of them.

The success of immunotherapy aimed at blocking immune checkpoints has changed the outlook on the treatment of selected adult cancers, namely, malignant melanoma, non-small cell lung cancer, Hodgkin lymphoma, clear-cell renal cell carcinoma, and others. Currently, six immunotherapeutics that block the key pathway of inhibiting the activity of T lymphocytes are approved by the FDA and registered by the EMA: anti-PD-1 (programmed death 1) monoclonal antibodies—nivolumab, pembrolizumab, and cemiplimab; and anti-PD-L1 (programmed death ligand 1) antibodies—atezolizumab, durvalumab, and avelumab. Such therapies function by augmenting existing anti-tumor T-cell responses that have been rendered ineffective by inhibitory pathways. The interaction of PD-1 and PD-L1 molecules leads to the extinction of the anti-cancer activity of T lymphocytes. The idea of immunotherapy using antibodies against immune checkpoints is based on blocking one of these molecules, which restores the cytotoxic activity of T lymphocytes [4].

The role of immune checkpoint inhibitors (ICIs) in the treatment of childhood cancers is still evolving. Single phase-I/II trials published in 2020–2022, where pembrolizumab (NCT0233266849), nivolumab (NCT0230445848), avelumab (NCT0345182551), and atezolizumab (NCT0254160450) were evaluated as monotherapies against recurrent and refractory childhood cancers, supply the most data. ICIs have been most advantageous in the therapy of classical Hodgkin lymphoma (HL) in children. As in adults, inhibitors of PD-1/PD-L1 have resulted in objective response rates of 30–60% in children and adolescents with relapsed HL. Similar effectiveness has not been observed in other types of pediatric cancers, where only about 3% of patients experienced an objective response. No response was observed in solid tumors, such as central nervous system tumors, neuroblastoma, rhabdomyosarcoma, Ewing sarcoma, and osteosarcoma. This ineffectiveness of the therapy probably represents a principal difference in tumor immunobiology in children compared to adults. Childhood tumors are commonly regarded as “cold cancers”, with a low mutational load and limited infiltrating T lymphocytes [4,5].

The aim of this nationwide study was to analyze the safety and effectiveness of ICIs used in pediatric patients with various types of highly advanced malignancies. We retrospectively verified individuals treated with ICIs in pediatric oncology centers in Poland between 2015–2023.

## 2. Methods

### 2.1. Study Design

All patients treated with IPIs for malignancies in Polish pediatric oncology centers between 15 December 2015 and 15 August 2023 were included to the study. The results of the pilot study were published in 2020 [6].

### 2.2. Treatment

Each analyzed patient received nivolumab or pembrolizumab at a dose consistent with the recommendations of the product characteristics. In 78.5% (33/42), anti-PD1 drug was administered as monotherapy: 29/33 patients were treated with nivolumab and 4/33 received pembrolizumab. In the remaining 21.5% (9/42) of patients, nivolumab was used in combination with ipilimumab (in 1 patient with malignant melanoma), dinutuximab beta (in 5 patients with neuroblastoma), and brentuximab vedotin (in 3 patients with HL). The applied doses of drugs were as follows: for nivolumab, 240 mg/dose IV every 2 weeks for individuals weighing ≥ 50 kg or 3 mg/kg/dose IV every 2 weeks for patients weighing < 50 kg; for pembrolizumab, 2 mg/kg/dose IV every 3 weeks (max 200 mg/dose). The median number of doses was 7.5 (range: 1–51).

### 2.3. Assessment of Therapeutic Response

Each of the analyzed patients underwent follow-up imaging tests every three months or when disease progression was suspected. Pseudoprogression has been defined as a reaction to immunotherapy manifested by tumor enlargement caused by the infiltration of tumor tissue by immune system cells. The incidence of pseudoprogression is approximately 10% in patients treated with ICIs [4,7]. The classical response evaluation criteria for solid tumors (RECIST) should not be used for immunotherapy; therefore, response criteria based on the immune response criteria (irRC) and immunotherapy RECIST (iRECIST) were adopted.

### 2.4. Adverse Events

All adverse events observed in individuals were reported using the Common Terminology Criteria for Adverse Events (CTCAE) versions 5.0 and 6.0.

### 2.5. Statistical Analysis

The primary endpoint of the study was the probability of overall survival (OS) determined by the Kaplan–Meier method, with comparison made by the log-rank test. Mean survival was determined by the Kaplan–Meier method. An event was defined as any death. The secondary endpoint was progression-free survival (PFS), with an event defined as any progression and death being the competing event. The Cox regression model was used in uni- and multivariate risk factor analysis for OS and PFS. The following factors were analyzed: age, primary diagnosis (Hodgkin lymphoma, skin melanoma, neuroblastoma, other malignancies), number of lines of previous therapy, and ECOG (Eastern Cooperative Oncology Group) performance status scale. Hazard ratio (HR) was calculated with a 95% confidence interval (95% CI) for each risk factor. Statistical significance was defined as *p* < 0.05. All the tests were two-sided. SPSS 29 (PS Imago Pro 9.0; IBM, Armonk, NY, USA) statistical package was used.

## 3. Results

### 3.1. Demographics

A total of 42 patients were qualified for the study, including 28 males (66.7%) and 14 females (33.3%), with a median age of 13.6 years (range = 0.1–17.9), treated with ICIs in 11 pediatric oncology centers in Poland. The indications for treatment with anti-PD1 as monotherapy or in combination with another immunotherapeutic agent were as follows: Hodgkin lymphoma (n = 11; all nodular sclerosis type); malignant skin melanoma (n = 9); neuroblastoma (n = 8); and other malignancies (n = 14), including osteosarcoma in two, brain tumor in two, renal cell carcinoma in two, and single patients with primary malignant melanoma of the brain, chondrosarcoma, hepatoblastoma, ovarian germ cell tumor, adrenal cortex tumor, mediastinal gray zone lymphoma, rhabdoid tumor of the liver, primary rhabdomyosarcoma of pancreas, and poorly differentiated gastric adenocarcinoma. Table 1, Table 2, Table 3 and Table 4 present the characteristics of patients depending on the underlying disease, respectively. Table 1: cutaneous malignant melanoma; Table 2: Hodgkin lymphoma; Table 3: neuroblastoma; Table 4: other malignancies. Among 14/42 patients, the *V600* mutation status in the *BRAF* gene was assessed—1/14 had the mutation detected.

### 3.2. Clinical Course

The median follow-up of all patients included in the study was 1 year. At the end of follow-up, complete remission (CR) was observed in 37.7% (15/42) or children, and disease stabilization (SD) in 9.5% (4/42), with a mean survival 3.6 (95% CI = 2.6–4.6) years. Individuals who survived (CR or SD) received immunotherapy as a median 3 (range = 1–6) line of therapy. Similarly, patients with disease progression (PD) were treated with ICIs as a median 3 (range = 1–10) line of treatment. In six patients, immunotherapy was still ongoing at the end of the study. The general condition of patients assessed during the administration of the first dose of ICI was worse in the children among whom progression occurred. The median ECOG score for individuals with CR or SD was 1 (range = 0–2), while for patients with PD, it was 2 (range = 0–4). The median number of ICI doses for children with CR, SD, and PD was 26 (range = 7–51), 15 (range = 1–20), and 7 (range = 3–21), respectively.

### 3.3. Progression-Free Survival

Probability of PFS at 3 years for all patients was 0.45 (Figure 1a). The best prognosis (PFS = 0.78) was observed in the group with malignant melanoma of the skin. For Hodgkin lymphoma, neuroblastoma, and other cancers, the PFS values were 0.65, 0.25, and 0.25 (*p* = 0.003), retrospectively (Figure 2a). In one patient with Hodgkin lymphoma, progression was observed while using ICIs, but death occurred a few months after nivolumab was discontinued during the next line of treatment. Significantly better effects of immunotherapy were confirmed in older patients (≥14 years of age) compared to children <14 years of age (PFS = 0.68 vs. PFS = 0.29; *p* = 0.012) (Figure 2b). Patients with ECOG ≤ 1 had PFS = 0.59, while individuals with ECOG ≥ 2 had PFS = 0.23 (*p* = 0.002) (Figure 2c). Individuals without previous lines of therapy presented a trend for better prognosis than those with ≥1 line of therapy (PFS = 0.86 vs. PFS = 0.38; *p* = 0.061) (Figure 2d). In multivariate analysis, diagnoses other than cutaneous malignant melanoma and Hodgkin lymphoma (HR = 5.5, *p* = 0.027) were unfavorable prognostic factors for PFS (Table 5).

### 3.4. Overall Survival

Probability of OS for the entire group at 24 months was 0.55 (Figure 1b). The best survival (OS = 1.0) was observed in the group of patients with Hodgkin lymphoma. For malignant melanoma of the skin, neuroblastoma, and other rare cancers, the OS values were, respectively, 0.78, 0.33, and 0.25 (*p* = 0.002) (Figure 3a). Significantly better effects of immunotherapy were confirmed in older patients (≥14 years of age) compared to children <14 years of age (OS = 0.95 vs. OS = 0.3, *p* < 0.001) (Figure 3b). Patients with ECOG ≤ 1 had OS = 0.74, while individuals with ECOG ≥ 2 had OS = 0.2 (*p* = 0.001) (Figure 3c). Individuals without previous lines of therapy presented a trend for better survival than those with ≥1 line of therapy (OS = 0.86 vs. 0.49; *p* = 0.165) (Figure 3d). In multivariate analysis, age <14 years (HR = 11.1; *p* = 0.044) and ECOG score > 1 (HR = 4.2; *p* = 0.036) were adverse prognostic factors for overall survival (Table 6). Effects of treatment in relation to diagnosis and division into age groups ≥14 and <14 years are presented in Table 7.

### 3.5. Safety

Adverse events were observed in 30.9% (13/42) patients, presenting as hypothyroidism (3/42), pneumonia (3/42), hyperthyroidism (2/42) acute pancreatitis (2/42), skin rash (1), diarrhea (1/42), blood hypotension (1/42), SIRS (1/42), arthritis (1/42), and hepatotoxicity (1/42). Serious adverse events accounted for 16.7% (7/42) of all patients. Pseudoprogression was observed in 7.1% (3/42) patients. One of the two patients on combination therapy with ipilimumab presented hypothyroidism, diarrhea, and hepatotoxicity. One person from the entire group (1/42) required discontinuation of treatment due to unacceptable side effects in the form of recurrent fever accompanied by high parameters of inflammation of unclear etiology and severe arthritis. The situation concerned a 7-year-old girl treated for recurrent metastatic cutaneous malignant melanoma. After seven doses of pembrolizumab, lasting remission has been observed for >5 years. In other patients among whom side effects occurred, they were relatively mild, were manageable by delaying treatment for several weeks and/or steroid administration and did not require discontinuation of therapy.

## 4. Discussion

Davis et al. presented the results of a multicenter phase-I–II study conducted in 23 hospitals in the USA, wherein they assessed the effectiveness of nivolumab in 72 patients, aged 1–30 years, diagnosed with the following cancers: melanoma, neuroblastoma, rhabdomyosarcoma, osteosarcoma, Ewing sarcoma, Hodgkin lymphoma, and non-Hodgkin lymphoma (NCT02304458). Responses were observed in individuals with Hodgkin lymphoma (30%; 3/10) and with non-Hodgkin lymphoma (10%; 1/10). All responders with objective responses (OR) had PD-L1 expression confirmed. OR were not observed in other tumor types. Mild toxicities were observed in most patients. The most common overall toxicities were anemia (47%) and fatigue (37%). Dose-limiting toxicities were present in 7% of patients [8].

In a study reported by Davis et al., the effectiveness of ICIs in the treatment of cutaneous malignant melanoma has not been demonstrated, but it should be noted that only one analyzed patient (1/70) was diagnosed with melanoma [8]. Single clinical trials assessing the effects of immunotherapy and other drugs in the treatment of malignant melanoma in patients under 17 years of age failed due to low enrollment (NCT01696045, NCT01519323). Therefore, there are limited options for adjuvant therapy among pediatric patients with advanced melanoma [9]. We included nine patients with cutaneous malignant melanoma in our analysis. In seven/nine cases where anti-PD1 drugs were used as monotherapy, CR was achieved with a median OS of 30.6 months. Most of them were patients > 14 years of age for whom the biology of the tumors is probably proportional to that in the adult population, where perhaps the success achieved can be seen.

KEYNOTE-051 was a phase-I/II open-label trial where 154 children with melanoma or a PD-L1-positive, refractory, or relapsed solid tumor or lymphoma were recruited (NCT02332668). All participants, with the median age 13 years (range = 8–15), were treated with pembrolizumab. In 45% (69/154) of patients, grade 3–5 adverse events were observed, most commonly anemia (9%) and lymphopenia (6%); 3% (4/154) of individuals discontinued treatment due to serious adverse events; and 1% (2/154) of patients died (1 due to pleural effusion and pneumonitis and 1 due to pulmonary oedema). Of 15 children with refractory or relapsed Hodgkin lymphoma, 2 had complete responses (CR) and 7 had partial responses (PR). Thus, 60% (9/15) patients achieved an objective response (95% CI 32.3–83.7). Of 136 individuals with solid tumors and other lymphomas, 8 had partial responses; the diagnoses in these participants were adrenocortical carcinoma, mesothelioma, malignant ganglioglioma, epithelioid sarcoma, lymphoepithelial carcinoma, and malignant rhabdoid tumor. The proportion of children with an objective response was 5.9% (95% CI 2.6–11.3) [10,11].

Front-line anthracycline-based chemotherapy platforms are associated with the most robust outcome for patients with classical Hodgkin lymphoma [12]. As salvage therapy, the combination of brentuximab vedotin plus bendamustin is a valid option for patients with c-HL with age ≥ 18 years [13]. CheckMate 744 was a phase-II study, where 44 patients with a median age 16 years (range = 5–30) with relapsed/refractory classical Hodgkin lymphoma without complete metabolic response before autologous hematopoietic cell transplantation (auto-HCT) were evaluated for a risk-stratified, response-adapted approach with nivolumab plus brentuximab vedotin (NCT02927769). Patients received four induction cycles with nivolumab plus brentuximab vedotin; those without complete metabolic response (Deauville score > 3) received brentuximab vedotin plus bendamustine intensification. Complete metabolic response rate was observed in 59% of individuals after induction with nivolumab plus brentuximab vedotin [14,15,16]. Sun et al. performed a systematic review and meta-analysis of 20 prospective studies assessing the effectiveness and safety of PD-1 and PD-L1 inhibitors in relapsed and refractory Hodgkin lymphoma. The authors analyzed a total of 20 studies involving 1440 adult patients of which 19 studies concerned treatment with PD-1 inhibitors and 1 study with a PD-L1 inhibitor. The pooled ORR for the 19 studies in which it was an indicator of final effectiveness was 79% (95% CI 73–85). The pooled CR rate for the 20 studies was 44% (95% CI 34–55). PR was analyzed in the 19 included studies, and the pooled PR rate was 34% (95% CI 26–42) [5,17]. The conclusions from the available literature are consistent with the treatment results achieved in our study group among individuals with relapsed and refractory Hodgkin lymphoma. There is a high probability that the observation related to a better treatment effect in the group of older patients (>14 years of age) correlates with histopathological diagnoses more typical for this age (nodular sclerosis Hodgkin lymphoma and malignant melanoma of the skin).

Neuroblastoma is the most common extracranial solid tumor in children, and about 50% of patients have metastatic or refractory disease. The prognosis has partially improved after adding the anti-disialoganglioside antibody dinutuximab beta to the multimodal therapy, but it is still unsatisfactory. Preclinical studies in mouse models showed that dinutuximab beta resulted in an upregulation of the PD-1 checkpoint in neuroblastoma cell lines, and combined treatment with this antibody and an anti-PD1 drug in the tested mice showed a synergistic effect [18,19]. Ehlert et al. presented two cases of patients, a 4-year-old female and 17-year-old male with refractory metastatic neuroblastoma, for whom the combination of dinutuximab beta with nivolumab led to a complete and a very good partial remission [20]. The effects of ICIs in our group of patients with neuroblastoma were poor, although complete remission was achieved in two/eight individuals. The older age of the patients (untypical for neuroblastoma) who responded well to treatment is noteworthy.

Cacciotti et al. conducted a retrospective review of data on 11 pediatric patients with relapsed or refractory CNS tumors treated with ICIs at Boston Children’s Hospital in 2018–2019. Diagnoses included high-grade glioma (n = 5), diffuse intrinsic pontine glioma (DIPG) (n = 2), craniopharyngioma (n = 1), ependymoma (n = 1), non-germinomatous germ cell tumor (n = 1), and high-grade neuroepithelial tumor (n = 1). In the entire group of patients, nine participants received combination therapy with ipilimumab and nivolumab, and two patients received pembrolizumab or nivolumab as monotherapy. All individuals had previously undergone radiotherapy. The median duration of immunotherapy was 6.1 months (range = 1–25). In seven patients, therapy was discontinued due to disease progression, and in two participants, unacceptable toxicities occurred (hypertransaminasemia and colitis). Responses observed included partial response (n = 3), stable disease (n = 7), and progressive disease (n = 1), with a durable response observed in two patients [21].

In another retrospective analysis, treatment with nivolumab outcome was assessed. The study group consisted of 10 children, aged 2 to 17 years, with recurrent or refractory brain tumors like high-grade glioma (n = 5), low-grade glioma (n = 1), pineoblastoma (n = 1), medulloblastoma (n = 1), ependymoma (n = 1), and CNS embryonal tumor (n = 1) treated at Rady Children’s Hospital San Diego from 2015 to 2017. In three cases (two with high-grade glioma and one with CNS embryonal tumor), a partial response to treatment at the primary tumor site was observed. Median survival for PD-L1-positive individuals was 13.7 weeks versus 4.2 weeks for PD-L1-negative patients (ρ = 0.08). The authors suggest that the use of ICIs in pediatric brain tumor patients should be limited to those with elevated PD-L1 expression [22].

The use of ICIs as a last-chance treatment in our patients with other advanced cancers (Table 4) was usually supported by single case reports or studies on small groups of patients in which this type of treatment proved effective [23,24,25,26,27,28,29,30,31,32,33]. We acknowledge several limitations in our study, including the retrospective nature of the study, the diversity of the types of diseases, and the age of the patients in the cohort group. However, our data contribute significant information that might aid in the development of prospective validation studies.

Adverse events were reported in 30.9% (13/42) of patients in our study group, of which just over half (7/13) had CTCAE grade 3/4 toxicities. Compared to the above-mentioned prospective studies using ICIs in pediatric patients [8,10], the frequency of side effects in our patients was significantly lower. In four clinical trials, where nivolumab (NCT0230445848), pembrolizumab (NCT0233266849), atezolizumab (NCT0254160450), and avelumab (NCT0345182551) were tested as monotherapies against refractory and recurrent pediatric malignancies, usually mild systemic effects (fatigue and fever, grade 1/2) were observed, and the most common immune-related adverse event was hepatic toxicity (elevated transaminases, grade 1/2) [4]. There is a risk associated with the retrospective nature of our analysis, and it is possible that mild side effects such as fatigue, fever, anemia, or elevated liver function tests that did not require therapeutic interventions were missed. It should be emphasized that both our data and the data collected from the experience of other authors indicate that ICIs are well tolerated in the group of pediatric patients, and severe immunological toxicities are observed significantly less often than in adults. However, this is not necessarily a positive. Numerous data suggest that the increased occurrence of immunological toxicities correlates with the effectiveness of the therapy [34]. We observed this phenomenon in one of our patients with recurrent melanoma, in whom after seven doses of pembrolizumab, immunological side effects in the form of arthritis and SIRS forced the premature termination of immunotherapy. Despite significantly shorter treatment than planned, the child achieved 5-year PFS.

## 5. Conclusions

ICIs are a valuable treatment option for pediatric patients diagnosed with melanoma or relapsed/refractory Hodgkin lymphoma, with nivolumab being most often used. The effectiveness of anti-PD1/PD-L1 drugs has not been shown for other types of lymphomas and solid tumors in children. Pediatric cancer immunobiology remains an area of active research seeking to better understand the optimal use of immunotherapy in these patients.

## Figures and Tables

**Figure 1 cancers-16-00968-f001:**
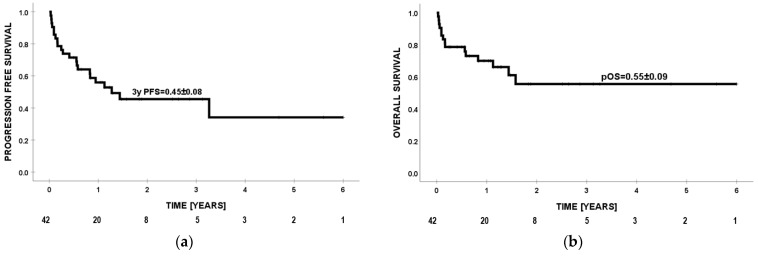
Progression-free survival (PFS) (**a**) and overall survival (OS) (**b**) of 42 patients from the first dose of anti-PD1 drug.

**Figure 2 cancers-16-00968-f002:**
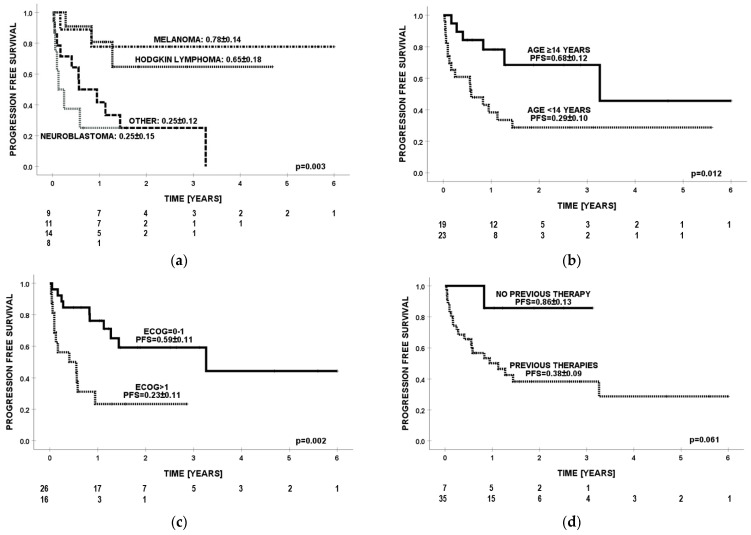
Progression-free survival (PFS) by (**a**) diagnosis, (**b**) age, (**c**) ECOG score, and (**d**) number of previous lines of therapies.

**Figure 3 cancers-16-00968-f003:**
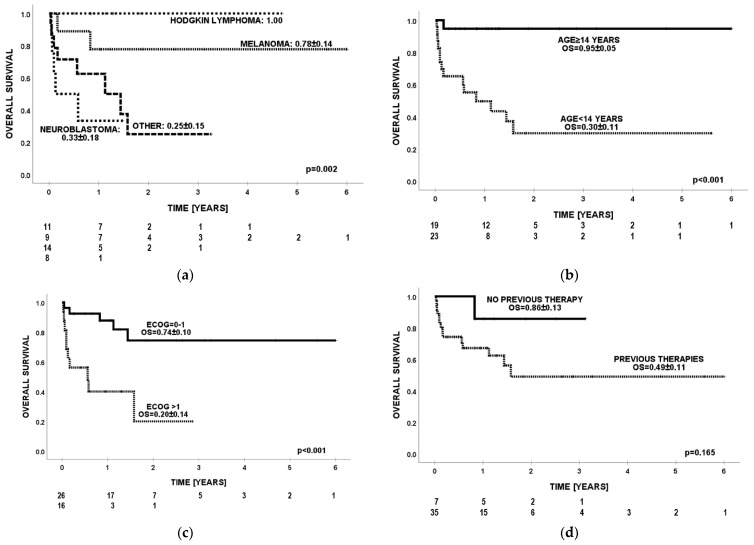
Overall survival (OS) by (**a**) diagnosis, (**b**) age, (**c**) ECOG score, and (**d**) number of previous lines of therapies.

**Table 1 cancers-16-00968-t001:** Characteristics of patients diagnosed with malignant melanoma of the skin.

No	Age [Years], Sex	ECOG Status	TNM Staging	Treatment (Number of Cycles)	Previous Lines of Therapy	Effect of Treatment	Adverse Events (CTCAE Grade)
**1**	16.5/M	0	IV	NIVO (27)	2	**CR (OS = 74.6 months)**	None
**2**	7.4/F	0	IV	PEMBRO (7)	2	**CR (OS = 68.1 months)**	SIRS (4); arthritis (4) *
**3**	8.8/F	0	III	NIVO (12)	0	**CR (OS = 38.1 months)**	None
**4**	15.7/F	0	III	NIVO (50)	0	**CR (OS = 30.6 months)**	Hypothyroidism (4); blood hypotension (3); diarrhea (2); tumor pseudoprogression
**5**	17.8/M	0	III	NIVO (48)	0	**CR (OS = 22.9 months)**	None
**6**	14.4/M	1	III	NIVO (27)	0	**CR (OS = 14.9 months)**	None
**7**	16.4/M	1	III	NIVO (26)	0	**CR (OS = 12.6 months)**	None
**8**	8.5/M	0	III	NIVO (9) + IPI (4)	0	PD	Hepatotoxicity (2)
**9**	3.1/F	1	IV	NIVO (4)	2	PD	Pneumonia (3)
**CUMULATIVE**	Median age: 14.4 years (M = 5; F = 4)	Good overall condition (ECOG > 2) in 9/9 patients	High stage of the cancer in 9/9 patients	Median ICIs doses: 26	Median: 0	CR in seven patients with median OS: 30.6 months	Good tolerance of therapy in most patients; * in 1/9 patients, side effects were unacceptable and forced the termination of therapy

F: female; M: male; NIVO: nivolumab; PEMBRO: pembrolizumab; IPI: ipilimumab; ICIs: immune checkpoint inhibitors; CR: complete remission; OS: overall survival; PD: progressive disease; SIRS: systemic inflammatory response syndrome; * The only patient in the study group for whom toxicity required discontinuation of anti-PD1 therapy.

**Table 2 cancers-16-00968-t002:** Characteristics of patients diagnosed with Hodgkin lymphoma (all: nodular sclerosis subtype).

No	Age [Years], Sex	ECOG Status	Treatment (Number of Cycles)	Previous Lines of Therapy	Effect of Treatment	Adverse Events (CTCAE Grade)
**1**	15.9/M	1	NIVO (8)	2	**CR (OS = 57 months)**	None
**2**	16.6/M	2	NIVO (26)	3	**CR (OS = 34.8 months)**	None
**3**	16.1//M	0	NIVO (26)	5	**CR (OS = 22.3 months)**	Pseudoprogression
**4**	17.2/M	2	NIVO (7)	4	**CR (OS = 15.8 months)**	Acute pancreatitis (2)
**5**	17.5/M	0	NIVO (20) + BV (1)	1	**SD (OS = 13.8 months)**	Rash (2)
**6**	17.1/M	1	NIVO (1) + BV (1)	4	**SD (OS = 13.1 months)**	None
**7**	14.8/F	0	NIVO (18)	1	**SD (OS = 10.3 months)**	None
**8**	17.2/M	1	NIVO (12)	3	**SD (OS = 6.1 months)**	None
**9**	16.3/M	1	NIVO (11) + BV (7)	2	PD	None
**10**	16.1/F	0	NIVO (4)	2	PD	Hyperthyroidism (2)
**11**	14.3/M	0	NIVO (4)	3	PD	Acute pancreatitis (2)
**CUMULATIVE**	Median age: 16.3 years (M = 9; F = 2)	Good general condition (ECOG > 2) in 9/11 patients	Median ICIs doses: 11	Median: 3	CR or SD in eight patients with median OS: 14.8 months	Good tolerance of therapy in most patients; no patient required termination of therapy due to side effects

F: female; M: male; NIVO: nivolumab; BV: brentuximab vedotin; ICIs: immune checkpoint inhibitors; CR: complete remission; OS: overall survival; SD: disease stabilization; PD: progressive disease.

**Table 3 cancers-16-00968-t003:** Characteristics of patients diagnosed with high-risk (HR) neuroblastoma.

No	Age [Years], Sex	ECOG Status	Treatment (Number of Cycles)	Previous Lines of Therapy	Effect of Treatment	Adverse Events (CTCAE Grade)
**1**	12.3/F	1	NIVO (13) + DB	4	**CR (OS = 18 months)**	Pneumonia (2)
**2**	10.2/M	1	NIVO (16) + DB	1	**CR (OS = 7.9 months)**	None
**3**	10.5/M	0	NIVO (6)	4	PD	None
**4**	3.2/M	4	NIVO (2) + DB	2	PD	None
**5**	5.6/M	4	NIVO (3) + DB	2	PD	None
**6**	7.0/M	2	NIVO (3)	3	PD	None
**7**	3.7/M	4	NIVO (2)	2	PD	None
**8**	3.2/F	4	NIVO (2) + DB	2	PD	None
**CUMULATIVE**	Median age: 6.3 years (M = 6; F = 2)	Poor general condition (ECOG ≤ 2) in 5/8 patients	Median ICIs doses: 3	Median: 2	CR in two patients	Good tolerance of therapy in all patients

F: female; M: male; NIVO: nivolumab; DB: dinutuximab beta; ICIs: immune checkpoint inhibitors; CR: complete remission; OS: overall survival; PD: progressive disease.

**Table 4 cancers-16-00968-t004:** Characteristics of patient with other advanced cancers characteristics.

No	Age [Years], Sex	ECOG Status	Diagnosis	Treatment (Number of Cycles)	Previous Lines of Therapy	Effect of Treatment	Adverse Events (CTCAE Grade)
**1**	12.1/M	1	Alveolar RMS of pancreas	NIVO (51)	3	**CR (OS = 32.1 months)**	Hyperthyroidism (3)
**2**	17.8/M	2	Mediastinal gray zone lymphoma	NIVO (9)	4	**CR (OS = 7.3 months)**	None
**3**	15.4//M	2	Osteosarcoma	NIVO (27)	4	PD	None
**4**	16.8/M	2	Osteosarcoma	NIVO (6)	2	PD	Pseudoprogression
**5**	16.4/F	0	Renal cell carcinoma	NIVO (17)	2	PD	Hypothyroidism (2)
**6**	3.4/M	0	Hepatoblastoma	PEMBRO (10)	9	PD	Rash
**7**	5.3/M	2	DIPG of brain	NIVO (2)	1	PD	Pneumonia (3)
**8**	16.6/M	3	Medulloblastoma of cerebellum	NIVO (4)	1	PD	None
**9**	17.9/F	2	Extrasceletal chondrosarcoma	NIVO (2)	3	PD	None
**10**	0.7/F	4	Rhabdoid tumor of kidney	NIVO (2)	1	PD	None
**11**	1.5/M	4	Rhabdoid tumor of liver	NIVO (1)	1	PD	None
**12 ***	18.0/M	2	Gastric cancer with lymphoid stroma	PEMBRO (3)	1	PD	None
**13**	16.9/F	0	Adrenal cortex cancer	NIVO (3)	3	PD	None
**14**	11.9/F	0	Ovarian germ cell tumor	PEMBRO (1)	5	PD	None
**CUMULATIVE**	Median age: 15.9 years (M = 9; F = 5)	Poor general condition (ECOG ≤ 2) in 9/14 patients	Most of the above diagnoses are ultra-rare cancers for the pediatric population	Median ICIs doses: 3.5	Median: 3	CR in two patients	Good tolerance of therapy in most patients; no patient required termination of therapy due to side effects

* presence of a mutation in BRAF gene, F: female; M: male; RMS: rhabdomyosarcoma; DIPG: diffuse intrinsic pontine glioma; NIVO: nivolumab; PEMBRO: pembrolizumab; ICIs: immune checkpoint inhibitors; CR: complete remission; OS: overall survival; PD: progressive disease.

**Table 5 cancers-16-00968-t005:** Risk factor analysis for progression-free survival (PFS).

Characteristics		Univariate Analysis		Multivariate Analysis	
Factor	Description	HR (95% CI)	*p*	HR (95% CI)	*p*
**Age**	>14 years	1		1	
	<14 years	3.2 (1.2–8.1)	0.017	2.0 (0.5–8.0)	0.322
**ECOG score**	0–1	1		1	
	>1	3.8 (1.6–9.1)	0.003	2.2 (0.6–8.5)	0.242
**Previous therapy**	None	1			
	≥1 line	5.5 (0.7–41)	0.096		
**Diagnosis**	HL or MM	1		1	
	Other	5.5 (1.2–25)	0.027	5.5 (1.2–25)	0.027

**Table 6 cancers-16-00968-t006:** Risk factor analysis for overall survival (OS).

Characteristics		Univariate Analysis		Multivariate Analysis	
Factor	Description	HR (95% CI)	*p*	HR (95% CI)	*p*
**Age**	>14 years	1		1	
	<14 years	14.3 (2.0–103)	0.009	11.1 (1.1–102)	0.044
**ECOG score**	0–1	1		1	
	>1	5.6 (1.8–16.6)	0.002	4.2 (1.1–16.4)	0.036
**Previous therapy**	None	1			
	≥1 line	3.8 (0.5–29.0)	0.197		
**Diagnosis**	HL or MM	1		1	
	Other	9.3 (2.1–41.8)	0.003	2.3 (0.25–20)	0.459

**Table 7 cancers-16-00968-t007:** Effects of treatment in relation to diagnosis and division into age groups (≥14 and <14 years).

Age [Years]	CR	SD	PD
**≥14**	5/5 with malignant melanoma 4/11 with Hodgkin lymphoma 1/1 with mediastinal gray zone lymphoma	4/11 with Hodgkin lymphoma	3/11 with Hodgkin lymphoma 2/2 with osteosarcoma 1/1 with renal cell carcinoma 1/1 with medulloblastoma of cerebellum 1/1 with extraskeletal chondrosarcoma 1/1 with gastric cancer with lymphoid stroma 1/1 with adrenal cortex cancer
*TOTAL:*	**58.3%% (14/24)**		**41.7% (10/24)**
**<14**	2/4 with malignant melanoma 2/8 with neuroblastoma 1/1 with alveolar RMS of pancreas		2/4 with malignant melanoma 6/8 with neuroblastoma 1/1 with hepatoblastoma 1/1 with DIPG of brain 2/2 with rhabdoid tumor 1/1 with ovarian germ cell tumor
*TOTAL:*	**27.8% (5/18)**		**72.2% (13/18)**

CR: complete remission; OS: overall survival; PD: progressive disease; RMS: rhabdomyosarcoma; DIPG: diffuse intrinsic pontine glioma.

## Data Availability

The data presented in this study are available on request from the corresponding author. The data are not publicly available due to privacy restrictions.

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
