# Peer review of "Anti-PD-1 Therapy in Advanced Pediatric Malignancies in Nationwide Study: Good Outcome in Skin Melanoma and Hodgkin Lymphoma"

_cancers, 2024, doi:10.3390/cancers16050968_

Round 1

Reviewer 1 Report

Comments and Suggestions for Authors

This study describes the experience with immune checkpoint inhibition (ICI) treatment for a variety of childhood cancers over a period of 8 years. Survival was variable as expected for such different tumor types. Severe adverse events were reported in 30.9% (13/42) patients.

Given the large variation in tumor types and their essentially different a priori expected outcomes, outcome results may not be of major relevance in this study, but tolerability for this age group is. Unfortunately, this aspect is only superficially evaluated in the results section and the toxicity spectrum in the present study is not discussed in light of previously reported studies in children or adults. Other studies are mentioned, but not discussed or explained in relation to the present results.

ICI treatment was first line in 7 melanoma patients, all others were treated in 2nd line or later (up to 10th line)Treatment response was largely most successful in melanoma and CHL patients as expected from results in adults. Various predictive factors are as expected (ECOG, #line of treatment, Dx). Better outcome in patients >14 years might at first sight be somewhat unexpected, but seems to be the result of significant enrichment for CHL/MGZL and melanoma patients and underrepresentation in high risk neuroblastoma patients. This notion is supported by the fact that age loses significance over diagnosis in a multivariate analysis. It is advisable to present all data split out per age group (</>14 years) in all table and discuss.

Please add #patients at risk per time point under the X-axis of all Kaplan-Meyer plots so that the sample size is clear (plateaus may only include one patient and the reader should be aware).

Discussion, line 254: does this refer to the same study by Davis and coworkers as mentioned above? Please clarify

Comments on the Quality of English Language

The English and level of writing are sufficient

Author Response

This study describes the experience with immune checkpoint inhibition (ICI) treatment for a variety of childhood cancers over a period of 8 years. Survival was variable as expected for such different tumor types. Severe adverse events were reported in 30.9% (13/42) patients.

Given the large variation in tumor types and their essentially different a priori expected outcomes, outcome results may not be of major relevance in this study, but tolerability for this age group is. Unfortunately, this aspect is only superficially evaluated in the results section and the toxicity spectrum in the present study is not discussed in light of previously reported studies in children or adults. Other studies are mentioned, but not discussed or explained in relation to the present results.

ANSWER: Information on the occurrence of side effects in the study group has been expanded in the revised version of the manuscript.

ICI treatment was first line in 7 melanoma patients, all others were treated in 2nd line or later (up to 10th line). Treatment response was largely successful in melanoma and CHL patients as expected from results in adults. Various predictive factors are as expected (ECOG, #line of treatment, Dx). Better outcome in patients >14 years might at first sight be somewhat unexpected, but seems to be the result of significant enrichment for CHL/MGZL and melanoma patients and underrepresentation in high risk neuroblastoma patients. This notion is supported by the fact that age loses significance over diagnosis in a multivariate analysis. It is advisable to present all data split out per age group (</>14 years) in all table and discuss.

ANSWER: According to this suggestion, all data split out per age group (</>14 years) in a separate new Table 7 were added and this issue has been discussed in the "Discussion" section.

Please add #patients at risk per time point under the X-axis of all Kaplan-Meyer plots so that the sample size is clear (plateaus may only include one patient and the reader should be aware).

ANSWER: All figures were modified according to the Reviewer's suggestion.

Discussion, line 254: does this refer to the same study by Davis and coworkers as mentioned above? Please clarify

ANSWER: Yes, the fragment mentioned by the Reviewer refers to Reference No. 8. The appropriate reference has been added as suggested.

Reviewer 2 Report

Comments and Suggestions for Authors

The manuscript by Marjanska et al presented their experience using immunotherapeutic agents in a wide variety of pediatric cancers.

The manuscript was well written. The authors presented the data clearly. Although I read the manuscript with great interest, I am not certain that new data were added to the literature. Immunotherapy has been shown to be effective in select pediatric cancer populations, namely those with melanoma and subsets of lymphomas. While the data presented do reinforce the existing experience, I am not certain new viewpoints are being presented.

Were the authors able to obtain data about the biology of the tumors that responded to therapy versus those that did not? If any of the tumors were sent for sequencing or delineation of tumor immune profiles, these might augment the manuscript.

Author Response

The manuscript by Marjanska et al presented their experience using immunotherapeutic agents in a wide variety of pediatric cancers.

The manuscript was well written. The authors presented the data clearly. Although I read the manuscript with great interest, I am not certain that new data were added to the literature. Immunotherapy has been shown to be effective in select pediatric cancer populations, namely those with melanoma and subsets of lymphomas. While the data presented do reinforce the existing experience, I am not certain new viewpoints are being presented.

ANSWER: Thank you for your review. The paper presents experience with immunotherapy in the treatment of advanced cancers in children collected from all pediatric oncology centers in Poland. It seems that in light of the lack of knowledge regarding the treatment of childhood cancers, resulting from the fact that most of them are rare diseases, new real-world data presented in the medical literature are valuable. We believe this explanation is sufficient and does not require any changes in the manuscript.

Were the authors able to obtain data about the biology of the tumors that responded to therapy versus those that did not? If any of the tumors were sent for sequencing or delineation of tumor immune profiles, these might augment the manuscript.

ANSWER: We regret, but only selected patients had mutations in the BRAF gene detected. Two patients with mutations in the BRAF gene are marked in the Tables (patient #7 in Table 2 and patient #12 in Table 4). Also, the presence of PD-L1 or SMARCB1 in cancer cells were not determined, and no other additional immunological or genetic parameters were analyzed. We believe this explanation is sufficient and does not require any changes in the manuscript.

Reviewer 3 Report

Comments and Suggestions for Authors

I commend the authors on conducting and presenting their study thoroughly and clearly. However, I would like to express my concerns regarding the ethical considerations of the research. I noticed that in the manuscript, there is no mention of ethical or Institutional Review Board (IRB) approval. Furthermore, in the declarations section, the authors have stated: "Institutional Review Board Statement: Not applicable."

Author Response

ANSWER: The study was conducted in accordance with the Declaration of Helsinki, and approved by the Institutional Review Board of Collegium Medicum in Bydgoszcz (approval numbers KB570/2017, KB152/2019, KB458/2019, KB558/2019). Respective information was added in the revised manuscript.

Round 2

Reviewer 3 Report

Comments and Suggestions for Authors

The ethical considerations have been adequately addressed by the authors. I have no further comments and believe that the manuscript is ready for publication in the current form. 

Author Response

The ethical considerations have been adequately addressed by the authors. I have no further comments and believe that the manuscript is ready for publication in the current form.

RESPONSE: We thank the Reviewer for this opinion.